# Cardiac-Specific Expression of Cre Recombinase Leads to Age-Related Cardiac Dysfunction Associated with Tumor-like Growth of Atrial Cardiomyocyte and Ventricular Fibrosis and Ferroptosis

**DOI:** 10.3390/ijms24043094

**Published:** 2023-02-04

**Authors:** Zhongguang Li, Qinchun Duan, Ying Cui, Odell D. Jones, Danyang Shao, Jianfei Zhang, Yuru Gao, Xixi Cao, Shulin Wang, Jiali Li, Xinjuan Lei, Wei Zhang, Liyang Wang, Xin Zhou, Mengmeng Xu, Yingli Liu, Jianjie Ma, Xuehong Xu

**Affiliations:** 1Laboratory of Cell Biology, Genetics and Developmental Biology, Shaanxi Normal University College of Life Sciences, Xi’an 710062, China; 2University Laboratory Animal Resources (ULAR), University of Pennsylvania School of Medicine, Philadelphia, PA 19144, USA; 3Department of Pediatrics, Columbia University, New York, NY 10032, USA; 4Department of Surgery, Davis Heart and Lung Research Institute, Ohio State University School of Medicine, Columbus, OH 43210, USA

**Keywords:** cardiac-specific Cre, heart failure, atrial tumors, matrix metalloproteinases, calcium channel, myocardial intercalated discs, ferroptosis

## Abstract

Transgenic expression of Cre recombinase driven by a specific promoter is normally used to conditionally knockout a gene in a tissue- or cell-type-specific manner. In αMHC-Cre transgenic mouse model, expression of Cre recombinase is controlled by the myocardial-specific α-myosin heavy chain (αMHC) promoter, which is commonly used to edit myocardial-specific genes. Toxic effects of Cre expression have been reported, including intro-chromosome rearrangements, micronuclei formation and other forms of DNA damage, and cardiomyopathy was observed in cardiac-specific Cre transgenic mice. However, mechanisms associated with Cardiotoxicity of Cre remain poorly understood. In our study, our data unveiled that αMHC-Cre mice developed arrhythmias and died after six months progressively, and none of them survived more than one year. Histopathological examination showed that αMHC-Cre mice had aberrant proliferation of tumor-like tissue in the atrial chamber extended from and vacuolation of ventricular myocytes. Furthermore, the αMHC-Cre mice developed severe cardiac interstitial and perivascular fibrosis, accompanied by significant increase of expression levels of MMP-2 and MMP-9 in the cardiac atrium and ventricular. Moreover, cardiac-specific expression of Cre led to disintegration of the intercalated disc, along with altered proteins expression of the disc and calcium-handling abnormality. Comprehensively, we identified that the ferroptosis signaling pathway is involved in heart failure caused by cardiac-specific expression of Cre, on which oxidative stress results in cytoplasmic vacuole accumulation of lipid peroxidation on the myocardial cell membrane. Taken together, these results revealed that cardiac-specific expression of Cre recombinase can lead to atrial mesenchymal tumor-like growth in the mice, which causes cardiac dysfunction, including cardiac fibrosis, reduction of the intercalated disc and cardiomyocytes ferroptosis at the age older than six months in mice. Our study suggests that αMHC-Cre mouse models are effective in young mice, but not in old mice. Researchers need to be particularly careful when using αMHC-Cre mouse model to interpret those phenotypic impacts of gene responses. As the Cre-associated cardiac pathology matched mostly to that of the patients, the model could also be employed for investigating age-related cardiac dysfunction.

## 1. Introduction

The Cre/LoxP system is a powerful tool to establish transgenic mouse lines to study gene function and generate animal models for human diseases, including cardiac diseases, cancer and many others [1,2,3,4]. Cre recombinase derived from bacteriophage P1 can catalyze homologous recombination between engineered loxP sites [5,6,7]. In the heart, inducible Cre-mediated excision of floxed genes driven by the cardiac-specific α-myosin heavy chain promoter (αMyHC-Cre) has been used extensively in conditional transgenic animal models [8,9,10,11]. αMHC-Cre has been shown to be cardiac myocyte-specific, which was used to drive efficient recombination of targeted genes [12]. However, accumulated evidence has revealed that sustained expression of Cre recombinase can be toxic [13,14]. Cre recombinase may induce possible intro-chromosome rearrangements, micronuclei formation and other forms of DNA damage independent of loxP transgenic sites in a variety of tissues and cells [15,16,17,18,19]. Cre recombinase-associated toxicities were reported in the heart of animal models on cardiac disease in multiple studies with less understood mechanism [20,21,22,23,24,25]. Moreover, in a widely used tamoxifen-sensitive Cre transgene under control of α-MHC promoter (αMHC-MerCreMer), Cre recombinase caused dose-dependent myocardial fibrosis, dilated cardiomyopathy (DCM), cardiomyocyte apoptosis and transient reduction of systolic cardiac function, leading to lethal heart failure [26,27]. Regulating concentration or activity of Cre recombinase by titrating the amount of tamoxifen can maximize recombination, while minimizing toxicity [27]. In addition, Cre recombinase could have potential for a variety of off-target effects. As endogenous, non-canonical loxP sites were targeted during designed Cre expression, activating DNA damage responses could associate with the cardiopathological pathway, which could promote gene expression leading to adverse myocardial remodeling [21]. However, the complex mechanisms and the risks associated with Cre-induced cardiotoxicity remain poorly understood.

As a form of regulated cell death, which is characterized by the iron-dependent accumulation of lipid hydroperoxides to lethal levels [28,29], ferroptosis is related to oxidative stress, inflammation, biosynthesis of glutathione, iron, amino acid and polyunsaturated fatty acid metabolism, which play an essential pathophysiological role in multiple diseases [28]. In recent years, studies found that ferroptosis is involved in the development of a variety of cardiovascular diseases (CVDs), including ischemia/reperfusion induced cardiomyopathy, doxorubicin-induced cardiotoxicity, atherosclerosis, myocardial infarction and heart failure [30,31,32,33,34]. Targeting ferroptosis is expected to become an effective therapeutic strategy for CVDs [30]. Whether ferroptosis participates in Cre-induced cardiotoxicity needs to be investigated.

In this study, we found that αMHC-Cre mice developed severe arrhythmias and cardiomyopathy and significant lethality beginning at six months of age, and the mice experienced abnormal atrial cell proliferation and atrial mesenchymal tumors. Our data showed that the cardiac function of αMHC-Cre mice displays a complex syndrome including atrial embolism, hyperplasia, cardiac fibrosis and reduction of the myocardial intercalated disc. Remarkably, data linked these cardiac dysfunctions to the ferroptosis-signaling pathway, with critical alternation of ferroptosis proteins, including PTGS2, Slc7a11 and 4-hydroxynonenal (4-HNE). Taken together, these findings suggest that cardiac-specific expression of Cre recombinase caused atrial cardiomyocyte proliferation, mesenchymal tumor-like growth, cardiac arrhythmia and heart failure, accompanied by structural changes of the intercalated disc, and ferroptosis plays an important role in Cre-mediated myocardial injury.

## 2. Results

### 2.1. Age-Related Progressive Death Accompanied with Arrhythmia and Heart Injury in αMHC-Cre Transgenic Mice

No apparent difference was observed in appearance outcome and physical behavior between the αMHC-Cre mice and wild-type mice until six months of age. Based on our systematic observation of their outcome and examination of their cardiac function characterized with ECG, no detectable phenotype can be perceived ranging from their birth to about six months. However, αMHC-Cre mice began to die after six months, and this lethality can reach up to 100% within approximately one year (*p* < 0.0001, Figure 1A), while the wild-type mice could live more than 400 days at least. To further understand and characterize the contribution of Cre recombinase to normal electrophysiological functioning in the heart, we performed surface ECGs on WT and αMHC-Cre mice at different stages. Surface ECGs recorded from anesthetized αMHC-Cre mice revealed arrhythmias and presented with atrioventricular (AV) conduction block at six months of age (Figure 1B). The RR intervals of αMHC-Cre mice followed a much wider distribution than WT littermates (Figure 1C,D). Heart rate in the αMHC-Cre mice was significantly lower than that in WT mice at the most (*p* < 0.0001, Figure 1E). Additionally, αMHC-Cre mice displayed significantly decreased P wave duration (*p* < 0.01) and significantly increased QRS duration (*p* < 0.01) and QT interval (*p* < 0.01), along with the same PQ interval (*p* < 0.05) compared to WT mice (Figure 1F). Then, we measured the serum levels of LDH and CK-MB, two biomarkers of heart failure. The documents displayed that these two cardiac enzymes, i.e., LDH (*p* < 0.001, Figure 1G) and CK-MB (*p* < 0.001, Figure 1H), were significantly higher in αMHC-Cre mice compared with WT mice at six months of age. The above analysis suggests that cardiac-specific expression of Cre recombinase can lead to cardiac failure, which is despondence for promotion of the mice’s premature death.

### 2.2. Atrial Tumor-like growth in αMHC-Cre Mice Led to Blockage of Blood Circulation and Heart Failure Bump

To identify the cause of mouse mortality and the effect of cardiac-specific expression of Cre recombinase on the above cardiac phenotype, dissection was applied on the αMHC-Cre mice, and we found that the dead mice had chest and abdominal effusion, abnormal heart enlargement and atrial congestion in particular compared to WT mice (Figure 2A). Interestingly, an extra tumor-like growth or lesions were in both atria of αMHC-Cre mice (Figure 2A). The ratios of heart weight to body weight (HW/BW) (*p* < 0.01) and heart weight to tibia length (HW/TL) (*p* < 0.05) were dramatically higher in αMHC-Cre mice than that in WT mice at the age of six months (Figure 2B). Histological analysis clearly exhibited that the atrium of αMHC-Cre mouse was full of these growths, which could disturb blood flow and even block it with a large mass of a tumor-like lesions (Figure 2C-a,b). These large tumor-like lesions were developed in either unilateral or bilateralatrium and eventually blocked blood circulation. Hematoxylin and eosin (H&E) staining showed that tumor-like lesions were composed of a large cell mass, which was clearly full of nucleus (Figure 2C-c,d), but not thrombus, as the other group reported [20]. This suggested that cells were dividing very rapidly, which was similar to the early development of tumor. Therefore, we detected the expression level of Ki67, which has been widely used as a proliferation marker for tumor cells [35]. As expected, Ki67 was significantly elevated in the atrium of αMHC-Cre mice by immunohistochemical staining (Figure 2D) and Western blot (Figure 2E).

Disturbed blood and even blocked flow generated with a large mass of a tumor-like lesions in αMHC-Cre atrium led to dysfunction of the pathological heart. We further inspected the ventricles of the αMHC-Cre heart and found that the left ventricular of αMHC-Cre mice showed severe myofiber cytoplasmic vacuolation (Figure 2F), which is associated with cardiotoxicity reflecting myofiber necrosis, fibrosis, mononuclear cell infiltration and etc. [36]. At the age of six months, these pathological changes caused by cardiac-specific expression of Cre recombinase were also accompanied by a significant increase in *Nppa* (*p* < 0.01), *Nppb* (*p* < 0.05) and *Myh7* (*p* < 0.05) mRNA levels in the left ventricle, which are three classic biomarkers of cardiac failure (Figure 2G).

Furthermore, we found that the mRNA level of apoptosis marker *Caspase3* was also significantly increased in the left ventricle of αMHC-Cre mice (*p* < 0.05, Figure 2H), along with left ventricle hypotrophy indicated with the size enlargement of left ventricular cardiomyocytes inspected with both Wheat germ agglutinin (WGA) staining (Figure 2I-a,b) and Periodic Schiff-Methenamine (PASM) staining (Figure 2I-c,d) measurement in αMHC-Cre mice (*p* < 0.001, Figure 2I,J). Together, these findings suggest that the atrium tumors-like growth leads to severe atrial congestion and impaired blood circulation, and subsequently, generates myocardial hypertrophy and myocardial failure eventually.

In order to comprehend the atrial tumor-like growths in αMHC-Cre mice at six months of age, we examined the expression levels of severity related proteins that played an important role in abnormal growth of tumorigenesis and development. As Epithelial-to-Mesenchymal Transition (EMT) is considered as crucial in tumorigenesis and development [37], along with two major hallmarks of EMT, E-cadherin and N-cadherin [38], we therefore examined the expression levels of these two proteins in the left atrium from WT and αMHC-Cre mice by Western blot at six months of age. Certainly, our results showed that the expression levels of both E-cadherin (*p* < 0.001) and N-cadherin (*p* < 0.0001) were significantly reduced in the left atrium of αMHC-Cre mice compared with WT mice (Figure 3A).

As MMP-2 and MMP-9 expressed in most all human cancers with a key role in degrading ECM and involving in tumor progression and metastasis [39], we inspected the expression level of MMP-2 and MMP-9 in the atrium with tumor-like growth. Our data showed that the expression level of MMP-2 was elevated approximately 2.5-fold in the left atrium of αMHC-Cre mice at six months of age compared to WT mice (*p* < 0.01, Figure 3B). On the other hand, MMP-9 expression was barely detectable in the left atrium of WT mice, whereas it was abundantly expressed in αMHC-Cre mice (*p* < 0.0001, Figure 3B). Meanwhile, the results of immune-histochemical staining showed that MMP-2 (Figure 3C) and MMP-9 (Figure 3D) were abundantly expressed in the left atrium tumor-like lesions of αMHC-Cre mice. Taken together, these data showed that mouse cardiac-specific expression of Cre recombinase leads to development of tumor-like lesions in the atrium, along with strong significant upregulation of MMP-2 and MMP-9 after six months of age.

### 2.3. α. MHC-Cre Mice Developed Severe Cardiac Fibrosis with Enhanced MMP-2 and MMP-9 Level

Cardiac fibrosis leading to heart failure and MMPs involved in the regulation of heart ECM proteins are well recognized in the cardiac pathogenesis process [40,41]. We subsequently inspected the characteristics of fibrosis in the heart of αMHC-Cre mice after six months of age before their death, as we presented above. Sirius red staining applied on heart sections showed that six-months-old hearts from αMHC-Cre mice exhibited severe fibrosis, which was predominantly in the left ventricle (*p* < 0.001, Figure 4A–C). In consistence with these staining results, the mRNA expression levels of myocardial fibrogenesis-related genes *Col1a1*, *Col3a1* and *Tgfb* in the left ventricle from αMHC-Cre mice were also significantly increased (*p* < 0.05, Figure 4D).

As extensive expression and accumulation of a collagen-rich ECM in fibrosis development is classically characterized as a signature for heart failure and arrhythmias, along with increased left ventricular stiffness [42], and MMPs exert a strong influence on cardiac fibrosis through multiple mechanisms [43], we further examined the signature with Western blots, along with immunohistochemistry analysis. Our results indicated that the expression levels of MMP-2 (*p* < 0.05) and MMP-9 (*p* < 0.01) proteins in the left ventricle from αMHC-Cre mice were significantly higher than that in WT mice (Figure 4E,F). More importantly, we found that the level of the MMP-9-activated form was also significantly increased (*p* < 0.01, Figure 4E,F). Additionally, the results of immunohistochemical staining showed that MMP-2 (Figure 4G-a,b) and MMP-9 (Figure 4G-c,d) were abundantly expressed in the left ventricle of αMHC-Cre mice (*p* < 0.05, Figure 4G,H). Together, our data suggest that Cre-induced cardiotoxicity is similar to classic cardiac fibrosis with the increase of MMP-2 and MMP-9 in active status.

The multiple studies involved with the higher count of participants concluded that the state of active myocardial remodeling is always positively related with enhanced activation of MMP-2 and MMP-9. This conclusion was also considered as the increase of these proteins in the circulating system of the patients [44]. In our case, to test the protein level of MMP-2 and MMP-9 in serum, we performed Western blot and found that significantly elevated levels of MMP-2 (*p* < 0.05) and MMP-9 (*p* < 0.01) proteins derived from αMHC-Cre mice compared to WT littermates at six months of age (Figure 5A,B). The protein level of the MMP-9-activated form was also significantly increased (*p* < 0.01, Figure 5B). Moreover, to compare the physiological relevance of MMP-2 and MMP-9 in heart failure between the animal model and humans, we analyzed the expression of *MMP-2* and *MMP-9* in the left ventricle tissue of donated patient samples using publicly available GEO datasets. Under idiopathic or ischemic heart failure condition, *MMP-2* was significantly elevated compared to the normal control (*n* = 16) in GSE5406, along with a trending increase in *MMP-9* mRNA expression due to limited numbers of the patients in this database (total number of idiopathic or ischemic heart failure is 194; idiopathic, *n* = 86 for; ischemic, *n* = 108) (*p* < 0.001, Figure 5C). In GSE57338, under the ischemic heart failure condition, *MMP-2* (*n* = 95) and *MMP-9* (*n* = 95) mRNA expression were significantly elevated compared to the normal control (*n* = 136) (*p* < 0.05, Figure 5D). Overall, our data suggest that the expression of Cre recombinase in the heart leads to cardiotoxicity associated with MMP-2 and MMP-9.

### 2.4. Alternation in Proteins Expression Associated with Intercalated Disc and Calcium Handling of αMHC-Cre Mice

To determine alternations of protein expression correlated with intercalated disc in Cre-induced heart injury response, Western blotting analysis was performed on the left ventricle from six-months-old WT and αMHC-Cre mice. Immunoblotting and its quantitation indicated that E-Cadherin and N-Cadherin, two protein components of the intercalated disc, mediated constant connection and normal physiological contraction of cardiomyocytes at the longitudinal ends of the myocardial cell membrane, which were significantly reduced in the left ventricle of αMHC-Cre mice (*p* < 0.05, Figure 6A,B). In contrast to E-Cadherin and N-Cadherin, β-Catenin expression was significantly upregulated in the αMHC-Cre myocardium (*p* < 0.05, Figure 6A,B). As immunofluorescence of N-Cadherin located in the intercalated disc (Figure 6C), compared with WT mice (Figure 6A-a–c), N-Cadherin of αMHC-Cre (Figure 6A-d–f) mice at the age of six months was dramatically decreased (*p* < 0.001, Figure 6D, right). The intercalated disks of αMHC-Cre hearts were irregularly shaped compared with the rectangular shape of control intercalated disks in the αMHC-Cre heart, indicated with an arrow, along with dissolving disc, indicated with arrows (Figure 6C). In consistence with previous publications [45,46,47], these data showed that cardiac-specific expression of Cre recombinase can reduce the intercalated disc of the myocardium in number per cardiomyocyte, and truncate their structure as well (*p* < 0.001, Figure 6D, left). As cardiac contractility is regulated by changes in intracellular calcium homeostasis and Ca^2+^ storge in endoplasmic reticulum (ER) [48] is mainly regulated by calcium channels and calcium pumps, including IP3R3, RyR and SERCA2 [49,50,51], we focused our targets on the expressions of these proteins. The results showed that the protein level of IP3R3 (*p* < 0.0001) was increased, and SERCA2 (*p* < 0.001) protein was decreased in the left ventricle from αMHC-Cre mice, although the protein level of RyR was not changed (*p* < 0.05, Figure 6E,F). Meanwhile, the expression of FKBP12, a regulatory protein of RyR, was significantly elevated, along with no change of another RyR regulatory protein, calcineurin A (CnA) (*p* < 0.001, Figure 6E,F). All above considered, these data suggest that cardiac-specific expression of Cre recombinase leads to pathological changes of the intercalated disc and furthermore could result in abnormality of the calcium-induced calcium-released signaling pathway in cardiomyocytes heading towards heart failure.

### 2.5. Ferroptosis Signaling Pathway Is Involved in Heart Failure Caused by Cardiac-Specific Expression of Cre Recombinase

As ferroptosis is recognized as newly discovered nonapoptotic cell death characterized by iron-dependent accumulation of lipid peroxides [28,29], many studies subsequently unveiled that ferroptosis plays an important role in the progression of heart disease [30,32,34]. To identify if the Cre recombinase promoted cardiac fibrosis and heart failure related to an underlying ferroptotic mechanism, we used real-time quantitative PCR to detect the mRNA expression level of *PTGS2* gene, a putative molecular marker of ferroptosis. *PTGS2* mRNA expression was significantly elevated in the left ventricle from αMHC-Cre mice compared with WT mice at the age of six months (*p* < 0.001, Figure 7A). Immunohistochemical staining showed that the expression of 4-HNE protein, which executes ferroptosis, with a reactive breakdown product of the lipid peroxides, was significantly increased (*p* < 0.05, Figure 7B,C). Western blot determined that the expression levels of PTGS2 (*p* < 0.01) and 4-HNE (*p* < 0.05) proteins were also significantly upregulated (Figure 7D,E). We further detected another ferroptotic marker, the expression levels of Slc7a11, which carry out inhibition on ferroptosis. Western analysis showed that expression level of Slc7a11 was significantly downregulated (*p* < 0.001, Figure 7D,E). All above included, these data suggest that cardiomyocytes ferroptosis promotes Cre recombinase-induced heart failure accompanied with the cardiac fibrosis, along with size increase of survival Cre recombinase overexpressed cardiomyocytes.

To further verify whether the related proteins in the animal model have the same expression pattern in human heart failure disease, we analyzed the expression levels of these proteins from the GSE5406 dataset. Compared with non-heart failure, the expression of *MMP-2*, *Col1a1*, *Col3a1*, *β-Catenin* and *PTGS2* was significantly higher (*p* < 0.05), and the expression of *SERCA2* was significantly lower (*p* < 0.001), while the expression of *MMP-9*, *E-Cadherin*, *N-Cadherin*, *CnA* and *RyR* was unchanged in idiopathic heart failure (Figure 7F). Furthermore, compared with non-heart failure, i.e., the expression of *MMP-2*, *Col1a1*, *Col3a1*, *N-Cadherin* and *PTGS2* was significantly higher (*p* < 0.01), and the expression of *SERCA2* (*p* < 0.05) was significantly lower, and the expression of *MMP-9*, *E-Cadherin*, *β-Catenin*, *CnA* and *RyR* was unchanged in ischemic heart failure (Figure 7F). Inclusively, the expression patterns of the most proteins (*MMP-2*, *Col1a1*, *Col3a1*, *N-Cadherin*, *β-Catenin*, *CnA*, *RyR*, *SERCA2* and *PTGS2*) in human heart failure are consistent with the discoveries above in the Cre recombinase-induced cardiotoxicity model.

It has been decades that cardiac-specific Cre for conditional knock-out gene of interest caused Cre recombinase cardiotoxicity. Our studies revealed a previously unknown mechanism of cardiotoxicity due to cardiac-specific Cre recombinase expression. The cardiac-specific expression of Cre recombinase resulted in atrial mesenchymal tumors, abnormal structural modifications of myocardial intercalated discs, featured with abnormal expression of N-Cadherin and E-Cadherin and protein. This Cre recombinase promoted cardiotoxicity also triggered cardiac interstitial and perivascular fibrosis characterized with abnormal expression of *Col1a1*, *Col3a1* plus *Tgfb* and MMP-2, and strong upregulated active-MMP-9. Altered expression of calcium-regulatory proteins IP3R3, SERCA2 and their regulator FKBP12 led to the cardiac dysfunction, and ferroptosis in the cardiac myocytes ultimately results in age-related mortality.

## 3. Discussion

Conditional gene editing based on inversion or excision of loxP-flanked DNA fragments by tissue-specific expression Cre recombinase is a powerful and standard technology for analysis of gene function in specific tissues [16,52,53]. Henceforth, multiple reports indicated that Cre recombinase exerted toxic effects in different types of cells and tissues, including extra defects in the heart [13,14,15,18,20,54,55,56,57]. In the present study, our data demonstrated that expression of Cre recombinase in cardiomyocytes showed serious unexpected effects on cardiac function. It caused heart injury, arrhythmia, atrial tumor, cardiomyocytes death and myocardial fibrosis, as we reported above. In contrast to previous studies [20,21,22,25,27], our current data revealed that cardiac-specific expression of Cre recombinase caused altered expression of intercalated disc-associated proteins and calcium-signaling-associated proteins, including downregulated N-Cadherin, E-Cadherin and SERCA2 and upregulated β-Catenin and IP3R3, which literally could lead to reduced cardiac contractile function, resulting in irreversible heart failure and the premature death of mice [58,59,60]. More importantly, our work discovered that activation of the ferroptosis signaling pathway, characterized with downregulated Slc7a11 and upregulated PTGS2 and 4-HNE, played an important role in Cre-mediated cardiotoxicity.

In cohort studies, some reports unveiled that men and women are in possession of different risks for cardiovascular diseases [61,62,63]. Generally, females are less likely to be identified the disease because of being prescribed with aspirin, statins, and certain blood pressure medications less often compared to males. A recent cohort study that collected data from 2020 and 2021 in the Jazan region unveiled that, among 498 coronary artery disease patients, 20.1 % were female (100 of 498) and 79.9 % were male (398 of 498) [61]. The genetic mechanism of the sex-bias in CVD was likely in possession of its significant molecular cause related to the FKBP506 binding protein 12.6 [64]. The deletion mice that experienced cardiac dysfunction with cardiac arrest [65,66] underwent sex-dependent CVD, which could be rescued with the cardiac-specific over-expression of FKBP12.6 [64]. However, in our study, there was no detectable different phenotype between the female and male αMHC-Cre mice. From the physiological perspective, the phenomenon may suggest that the hormone has no effect on the cardiomyopathy caused by Cre recombinase, which may be involved in the non-ryanodine receptor-associated calcium pathway because, in our case, ryanodine receptor was not regulated based on the amount of alternation of the RyR in Cre mice (Figure 6E,F).

Cardiac toxicity caused by cardiac-specific expression of Cre recombinase has been previously reported. Buerger et al. reported that mice with high-level myocardial expression of Cre recombinase developed dilated cardiomyopathy and premature death from congestive heart failure, accompanied by the increased expression of heart failure markers and cardiomyocyte apoptosis [20]. Using the same αMHC-Cre mice, Rehmani et al. found that αMHC-Cre mice showed symptoms of dilated cardiomyopathy at 7 months old, which eventually led to their complete death within a year [22]. Cre recombinase can activate DNA damage response through downregulation of activated p38 and increased expression of JNK, p53, and Bax, which induced myocyte death and heart fibrosis [22]. Using tamoxifen-sensitive Cre transgene under control of α-myosin-heavy-chain promoter, Bersell et al. found that expression of Cre recombinase in cardiomyocytes induced illegitimate DNA breaks, resulting in cardiomyocyte apoptosis, cardiac fibrosis and cardiac dysfunction, which were dependent on dose of tamoxifen applied [27]. Tamoxifen-induced cardiac-specific activation of Cre recombinase in mice caused associated significant hypertrophy and alterations in mitochondrial ATP and important proteins, which involved the regulation of cardiac oxidative phosphorylation [24]. Pugach et al. found that αMHC-Cre mice displayed cardiac toxicity with fibrosis, inflammation, and DNA damage in the heart and decreased cardiac function by six months [21]. Prolonged Cre expression can target endogenous, non-transgenic loxP sites, so as to activate DNA damage response related to pathological signaling pathway and adverse myocardial remodeling [21]. From our data presented above, we can find that Cre-mediated cardiomyopathy is age-dependent, and the Cre transgenic mice died after six months of age. Our result suggests that, using appropriate genetic control of Cre expression, the conducted experiments for the conditional knock-out of gene of interest before this age could conclude dependable results of well-designated experiments.

Many studies reported similar cardiomyopathy after cardiac gene deletion using Cre/LoxP system [67,68,69,70,71,72,73,74,75,76,77,78,79,80,81,82,83]. Whether the subsequent cardiac phenotype is due to the loss of the specific loxP-targeted gene or the effect of Cre activation is unclear, because Cre activated in mice in the heart was not selected as the control group in these studies. Researchers often overlook the cardiotoxicity caused by Cre recombinase, probably because the mice used for the experiments are usually 8–12 weeks old, a period before Cre has accumulated to a dose that produces cardiotoxicity. Furthermore, the LoxP sequence similar sites can cause Cre-mediated integration and excision in the genomes of mice [21]. Heart interleukin-4 (IL-4) receptor α defective mice, which were produced by Cre/loxP-mediated recombination system, displayed intercalated disk defects associated with dilated cardiomyopathy, atrial thrombosis and heart failure [84]. The phenotypes appear very similar to the phenotypes of αMHC-Cre mice that we and others reported. At the same time, the authors emphasized the importance of careful assessment and detection of unexpected adverse phenotypes, as they may not necessarily be correlated with targeted gene editing, but with spontaneous or unexpected mutations during strain creation [84].

Cardiac fibrosis is not only an important feature of cardiac remodeling, but also a hallmark of heart failure [85]. Excessive deposition of ECM in cardiac interstitium resulted in the increase of passive myocardial stiffness and the gradual deterioration of both systolic and diastolic function, and finally developed to lethal arrhythmia and heart failure [85]. As matrix metalloproteinases, MMPs are in charge of organizing ECM remodeling and degradation in ischemia-reperfusion induced heart failure and cardiovascular disease and cancers [86]. Alterations of MMPs activity regulation were believed to be important contributors to the progression of heart failure [87]. Our data showed that αMHC-Cre mice had age-related severe fibrosis after four months of age. In addition, compared with WT mice, expression levels of MMP-2 and MMP-9 in αMHC-Cre mice were significantly increased. Importantly, the activation level of MMP-9 was significantly increased, along with aging. These findings provide extra evidence for the increase of MMPs in heart failure caused by the cardiac overexpression of Cre recombinase.

Intercalated disks maintain cardiac tissue structural integrity and synchronized contraction [88]. Mutation or defect of ICD components, such as fascia adherens, desmosomes, and gap junctions, lead to poor myocardial contraction and stagnation of intracardiac blood, which in turn leads to dilatation of the heart and ultimately to the clinical symptoms of heart failure [88]. Our results showed that the expression of intercalated disc related proteins E-cadherin and N-cadherin in αMHC-Cre mice was decreased significantly, while the expression of β-catenin was increased significantly. Additionally, abnormal calcium homeostasis leads to electrical and systolic dysfunction, and can lead to dilated cardiomyopathy and heart failure [89,90,91,92]. We examined the expression levels of IP3R3, RyR2 and SERCA2, three major intracellular calcium-regulated proteins in αMHC-Cre mice. The results showed that expression of IP3R3 was significantly increased, and expression of SERCA2 was dramatically decreased, while expression of RyR2 was unchanged in the left ventricular of αMHC-Cre mice. These data suggest that αMHC-Cre expression resulted in abnormality of intracellular calcium homeostasis and perturbations in Ca^2+^ handling cardiomyocytes caused by Cre-mediated cardiotoxicity. Likewise, in vitro cellular level transfection of Cre recombinase is required to detect alternation in calcium ion levels to further validate the effect of Cre recombinase on calcium ion released from cardiomyocytes.

Generally, myocardial cell death is considered as the primary progression of cardiomyopathy, and previous studies showed that cardiomyocyte apoptosis is involved in Cre-mediated cardiotoxicity [20,21,22,27]. Since the accumulation of iron-dependent lipid peroxides is most likely recognized as the signature of ferroptosis [93], our data showed that ferroptosis was approved as an important manner of cardiomyocyte death in Cre-mediated cardiotoxicity. The expression level of ferroptosis marker PTGS2 was significantly increased in the hearts of αMHC-Cre mice at the age of six months. Further, we found that 4-HNE, a stable product of lipid peroxidation contributed to the cytotoxic effects of oxidative stress, was markedly increased as well. As the predominant ferroptosis defense system is the Slc7a11-glutathione-GPX4 (Slc7a11-GSH-GPX4) signaling axis, and inactivation of Slc7a11-GSH-GPX4 axis would like to generate an accumulation of lipid peroxides and subsequently leads to ferroptotic cell death [93,94,95,96], which suggests this manner of myocardial death in αMHC-Cre mice. Based on the discussion linking to our results, our data conclusively provide this alternative mechanism of myocyte death in αMHC-Cre mice. Sustained expression of exogenous Cre recombinase in mouse heart caused persistent oxidative stress, resulting in myocardial cell membrane lipid peroxidation with progressive severity of cardiac injury.

In human cardiovascular diseases, the CVD patients with developing heart failure suffer severe cardiac condition, in which the heart does not inflate enough blood to systematic circulation and pulmonary circulation leading to cardiac arrest. The complexity of CVD-associated heart failure needs more cellular and molecular comprehension to development more efficient strategy on treatment and diagnostics. The comprehensive understanding based on animal models combined with proficient database derived from fast increasing numbers of human patients appears a unique opportunity to expend current approach to eventually decrease the risk of death. According to our findings in this study, a couple of pathways may be considered as new targets to develop for heart failure, such as RyR- and IP3R-mediated calcium signaling, MMP-cadherin-associated cardiac architecture, including intercalated disks of cardiomyocytes, and collagen-marked cardiac fibrosis. In addition to these, we unveiled the Cre-promoted ferroptosis signaling indicated with the expression of increasing PTGS2/4-HNE and decreasing Slc7a11. This ferroptosis signaling could be considered as a unique target for developing a pharmacological approach against health challenges from heart failure.

Calcium-induced calcium release (CICR) mediated with RyR and IP3R, along with their regulators, such as FKBP12, is considered as the main pathway for excitation-contraction (EC) coupling in muscle cells including cardiomyocytes in the heart [51,97,98]. As the RyR regulator, an immunophilin protein FKBP1A can bind to immunosuppressive drugs, such as FK506, rapamycin and cyclosporin A (CsA), and interacts in the mTOR pathway, which are fundamental for CICR to maintain the EC coupling in general physiological condition. In our study, the significant increase of IP3R and FKBP12, but not RyR (Figure 6E,F), indicated that Cre-mediated cardiopathology did involve the regulation of RyR activity as a calcium channel in ER, although the RyR expression was not increased in the Cre cardiomyocytes. The related CICR calcium influx from the ER to cytosol could be compensated via the increased IP3R proteins (Figure 6E,F). This hypothesis of the IP3R CICR calcium influx needs more evidence in cardiomyocytes, as the interaction between IP3R and FKBP12 was proved in cardiomyocytes mediated by calcineurin expression in early 1995 [99] and no further research proved this interaction.

## 4. Materials and Methods

### 4.1. α. MHC-Cre Transgenic Strains and Animal Care

Transgenic founders (B6.FVB-Tg (Myh6-cre) 2182Mds/J) were purchased from Jackson Laboratory (Bar Harbor, ME, USA), which were donated by Michael Schneider from Imperial College London. They were housed in a standard controlled environment and allowed free access to water and food in a facility with a 12h light, 12h dark cycle. Animal care and experiments were performed in accordance with the protocol approved by the Animal Care and Use Committee of Shaanxi Normal University (ACUC-SNNU), and all manipulations were conducted in consistence with established guidelines.

### 4.2. α. MHC-Cre Transgenic Strains and Genotype Identification

The B6.FVB-Tg (Myh6-cre) 2182Mds/J founders with over expression of cardiac-specific α-myosin heavy chain promoter driven Cre recombinase (αMyHC-Cre) were used to breed for all experiments in this study. Age-matched wild-type (WT) littermates were used as controls in the study. Animal maintenance of both αMyHC-Cre mice and WT littermates were housed at room temperature in a range of 21–26 °C and in cycling of 12 h day-time/12 h night time and routine conditions in the protocol approved by the ACUC-SNNU. For mouse genotyping, genomic DNA was extracted from mouse tail using proteinase K (20 mg/mL; Cat No. 25530049, Thermal Fisher Scientific, Waltham, MA, USA) digestion method and genotyped representative of transgenic band in size of approximately 100 bp via polymerase chain reaction using Cre-specific primers including forward Cre F oIMR1084, GCGGTCTGGCAGTAAAAACTATC and reverse Cre R oIMR1085, GTGAAACAGCATTGCTG TCACTT control with internal positive of positive band in size of 324 bp using forward primer ATGACAGACAGATCCCTCCTATCTCC, and reverse primer CTCATCACTCGTTGCATCATC GAC. All primers were from manufactured designation (GenScript, Nanjing, China).

### 4.3. Electrocardiography (ECG)

Three-lead surface ECGs were measured by using limb electrodes, as previously described [11]. Briefly, before ECG examination, mice were lightly anesthetized with 20 mg/mL of avertin solution and then placed on a platform. Then, 20 mg/mL of avertin solution was prepared from 1.6 g/mL avertin stock solution (15.5 mL of 2-methyl-2-butanol (tert-amyl alcohol) in 25 g of avertin (2-2-2 tribromoethanol). Tert-amyl alcohol (Cat No. 8061931000) and 2-2-2 tribromoethanol (Cat No. T48402-25G) were purchased from Sigma (Sigma, St. Louis, MO, USA). A temperature-controlled thermal metallic insulator was used to maintain body temperature at 37 °C. ECG data were amplified and recorded with an amplifier (MD3000, Xuzhou Lihua Electronic Technology Development Co., Xuzhou, China). ECG signals were documented for 5 min for analysis after mice paws contacted with electrodes. Each recording was analyzed using the software of MD3000 Multi-channel electrophysiology recorder (Version: MD3000 v3.2). Heart rate (HR), P wave duration, PQ interval, QRS duration and QT interval were obtained by analyzing three stable ECG traces of 30 s in at least three different timing points to guarantee recording consistency.

### 4.4. Measurement of Serum LDH and CK-MB

After collection of whole blood, allow the blood to clot by leaving it undisturbed overnight at 4 °C. Remove the clot by centrifuging at 8,000 rpm for 20 min to obtain serum in a refrigerated centrifuge. Serum lactate dehydrogenase (LDH) levels were detected by using a LDH activity test kit (Cat No. D799208-0100, Sangon Biotech, Shanghai, China), and creatine kinase-MB (CK-MB) levels were measured using a mouse CK-MB ELISA kit (Cat No. ml037723, Enzyme-linked Biotechnology, Shanghai, China), in accordance with the manufacturer’s instructions.

LDH catalyzes the oxidation of lactic acid by NAD^+^ to produce pyruvate, and further interacts with 2, 4-dinitrophenylhydrazine to generate pyruvate dinitrophenylhydrazone, which shows brownish red color in alkaline solution, and this color shade is proportional for the concentration of pyruvate. Absorbance of all samples was measured at 450 nm in a 96-well plate using Synergy LX Multi-Mode Reader (SLXFA BioTek Instrument; Winooski, VT, USA). Serum LDH of animals is determined by comparing a standard curve according to each measured value at absorbance_450 nm_ to concentration of corresponding standard.

CK-MB ELISA Kit (Cat No. ml037723, Enzyme-linked Biotechnology, Shanghai, China) used for detecting serum creatine kinase-MB is based on a solid phase sandwich enzyme-linked immunosorbent assay (ELISA) using microtiter enzyme standard plate for detection with biotin-labeled antibody. Affinity-labeled HRP was applied for enzyme-specific conjugation, and then substrates A and B were added. The color was produced and the shade of color was proportional to the concentration of substances for each measured sample. The optical density value of each well was collected at 450 nm of wavelength using Synergy LX Multi-Mode Reader as the above.

### 4.5. Histological Evaluations, Immunohistochemistry and Immunofluorescence

Hearts were harvested and fixed overnight in 4% paraformaldehyde (pH 7.4) and then processed for paraffin embedding. Paraffin-embedded hearts were sectioned at thickness of 5 μm and subsequently stained with hematoxylin and eosin (H&E) for routine histologic examination or Sirius Red for cardiac fibrosis using a standard protocol. Images were obtained in bright field microscope or polarized light microscope. The Sirius Red positive area was quantified as a fraction of total tissue area under bright field microscopy by Image J (NIH, Bethesda, MD, USA). Immunofluorescent staining of N-Cadherin (1:100; Cat No. sc-7939, Santa Cruz Biotechnology, Inc., Dallas, TC, USA) was performed as follows: slides were deparaffinized and rehydrated by incubating successively in xylene, 100% ethanol, 95%, 85%, 70%, 50% ethanol, and phosphate-buffered saline (PBS). Antigen retrieval was achieved by heating in a pressure cooker with Tris-EDTA buffer (pH 9.0) for 15 min. The sections were blocked with PBS containing 3% (*w*/*v*) bovine serum albumin (BSA) (Cat No. A1933, Sigma, St. Louis, MO, USA) at room temperature for 1h, and then incubated with N-Cadherin at 4 °C overnight, followed by incubation for 1 h with Fluorescein (FITC) AffiniPure Goat Anti-Rabbit IgG (H + L) secondary antibody (1:500, Cat No. 111-095-003 Jackson ImmunoResearch Inc.; West Grove, PA, USA) at room temperature. DAPI was used to stain the nucleus. For immune-histochemistry staining, sections were blocked for endogenous peroxidases in 0.3% H_2_O_2_ for 10 min after antigen retrieval. The sections were blocked with PBS containing 3% (*w*/*v*) BSA at room temperature for 1h, and then incubated overnight with Ki67 (1:100; Cat No. ab15580, Abcam Cambridge, UK), MMP-2 (1:100; Cat No. ab37150, Abcam, Cambridge, UK), MMP-9 (1:100; Cat No. ab38898, Abcam, Cambridge, UK), or 4-HNE (1:100; Cat No. ab46545, Abcam, Cambridge, UK) at 4 °C under humidified conditions. Horseradish peroxidase-conjugated secondary antibodies (Dako EnVision HRP; Dako, Copenhagen, Denmark) were applied for 60 min and visualized with diaminobenzidine after an incubation for 2 min incubation at room temperature. Finally, the slides were counterstained with hematoxylin, dehydrated and covered. Images were captured on a Zeiss Observer A1 fluorescence microscope and analyzed by Image J (Version: ImageJ 1.53c).

### 4.6. Quantitative Real-Time PCR

Total RNA was extracted from heart tissue using TRIzol reagent (Cat No. 15596018, Invitrogen, Carlsbad, CA, USA), according to the manufacturer’s instruction. First, 1 μg of total RNA was reverse-transcribed to complementary DNA using the PrimeScript™ RT Master Mix kit (Cat No. RR036A, Takara, Osaka, Japan) in accordance with the manufacturer’s instructions. The reaction of PCR was performed with a CFX96 Real-Time System (Version: 4.1.2433.1219; Bio-Rad, Hercules, CA, USA) and TB Green Premix Ex Taq II (Tli RNaseH Plus) (Cat No. RR820A, Takara, Osaka, Japan) with specific primers. The following primers (5′-3′) were used. *GAPDH*: forward, AGGTCGGTGTGAACGGATTTG; reverse, TGTAGACCATGTAGTTGAGGTCA. *Caspase3*: forward, GAGCTTGGAACGGTACGCTA; reverse, GAGTCCACTGACTTGCTCCC. *Nppa*: forward, GAGGAGAAGATGCCGGTAGA; reverse, AGCCCTCAGTTTGCTTTT. *Nppb*: forward, GCCAGTCTCCAGAGCAATTC; reverse, TCCGATCCGGTCTATCTTGT. *Myh7*: forward, AGATGGCTGGTTTGGATGAG; reverse, CGCACTTTCTTCTCCTGCTC. *Col1a1*: forward, AGAGCATGACCGATGGATTC; reverse, CCTTCTTGAGGTTGCCAGTC. *Col3a1*: forward, CGTAAGCACTGGTGGACAGA; reverse, CGGCTGGAAAGAAGTCTGAG. *Tgfb*: forward, CAACAATTCCTGGCGATACC; reverse, GAACCCGTTGATGTCCACTT. *PTGS2*: forward, CTGCGCCTTTTCAAGGATGG; reverse, GGGGATACACCTCTCCA CCA. Real-time PCR reactions were amplified at 95 °C for 30 s and 60 °C for 30 s for 40 cycles, following an initial denaturation step at 95 °C for 3 min. Relative gene expression was calculated by 2^−ΔΔCT^ method, using *GAPDH* as a housekeeping control. Results were represented as fold change by normalizing to control groups.

### 4.7. Western Blot

Heart tissues were lysed by homogenizing in a RIPA buffer (150 mM NaCl, 10 mM Tris-HCl, pH 7.2, 0.5% SDS, 1% NP-40 and 0.5% deoxycolate) containing a cocktail of protease inhibitors (Cat No.C510009, Sangon Biotech). Equal amounts of proteins (20 µg) were separated by 8% or 12% SDS-PAGE (sodium dodecyl sulfate–polyacrylamide gel electrophoresis) and transferred to a polyvinylidene difluoride (PVDF) membrane. The membranes were blocked with 5% nonfat milk in Tris-buffered saline containing 0.05% Tween-20 (TBST) for 1h at room temperature and then probed with primary antibody (diluted in TBST supplemented with 5% BSA) overnight at 4 °C on an orbital shaker with gentle shaking. Membranes were then washed with TBST and incubated with the appropriate horseradish peroxidase–conjugated secondary antibody for 1h. Finally, membranes were developed with highly sensitive ECL luminescence regent (Cat No. C500044, Sangon Biotech, Shanghai, China) on an automatic chemiluminescence image analysis system. Densitometric analysis of blots was performed by Image J (NIH, Bethesda, MD, USA).

GAPDH, PTGS2 were purchased from Proteintch (Chicago, IL, USA); Ki67, MMP-2, MMP-9, Slc7a11 and 4-HNE, from Abcam (Cambridge, UK); N-Cadherin, β-Catenin were purchased from Santa Cruz Biotechnology (Dallas, TX, USA); E-Cadherin, CnA were purchased from Cell Signaling Technology (Danvers, MA, USA); RyR2 was purchased from Millipore (Burlington, MA, USA); IP3R3 was purchased from Novus (Littleton, CO, USA); high sensitive ECL luminescence regent was purchased from Sangon Biotech (Shanghai, China).

### 4.8. Analysis on Expression of MMP-2 and MMP-9 in Human Heart Failure

Publicly available Gene Expression Omnibus (GEO) datasets were downloaded. The GEO accession number used for analysis are GSE5406 (https://www.ncbi.nlm.nih.gov/geo/query/acc.cgi?acc=GSE5406, accessed on 10 December 2022) [100] and GSE57338 (https://www.ncbi.nlm.nih.gov/geo/query/acc.cgi?acc=GSE57338, accessed on 10 December 2022) [101]. The samples of heart left ventricle were used for their extracting of RNA based on their procedure. The mRNA expression data were normalized by log2 transformation and analyzed with GEO2R. In GSE5406 dataset: *n* (non-failing heart) = 16; *n* (Idiopathic failing heart) = 86; *n* (Ischemic failing heart) = 108. In GSE57338 dataset: *n* (non-failing heart) = 136; *n* (Idiopathic failing heart) = 82; *n* (Ischemic failing heart) = 95.

### 4.9. Statistical Analysis

All of the experimental data were carried out with GraphPad Prism Software v.7.0 (GraphPad Software Inc., San Diego, CA, USA). Data presented in dot plots were analyzed by two-tailed, unpaired Student’s *t*-test. Data were expressed as the mean ± standard error of the mean. A value of *p* < 0.05 was considered significant.

## 5. Conclusions

In this study, we unveiled that the cardiac overexpression of Cre recombinase directly causes atrial tumor-like growth (but not thrombus accumulation in its atrium) indicated with massive proliferated atrial cardiomyocytes characterized with tumor-specific protein Ki67 in the Cre recombinase mice. The cell massiveness could block the circulating flow through atria with less blood injection and generates ventricle pathological issues associated with ventricle-expressed Cre recombinase. The cardiac-specific Cre recombinase leads to cardiac hypotrophy with significant enlargement, which is accompanied by cardiomyocyte structural defect indicated with irregular expression of the intercalated disc-specific proteins N-Cadherin and E-Cadherin. The disturbed calcium signaling pathway was indicated with dramatically increased expression of IP3R3 plus its possible regulator FKBP12, and intensely decreased expression of SERCA2 cannot support cardiomyocytes performing their normal physiological duty in the mice. Meanwhile, the cardiac fibrosis is also promoted with the featured phenotype of interstitial and perivascular fibrosis, signatured with upregulated *Col1a1*, *Col3a1* and *Tgfb* expression, along with a significant increase of MMP-2 and surge upregulated active-MMP-9. Eventually, the cardiac expression of Cre recombinase promotes cardiac ferroptosis characterized with intensive upregulated expression of PTGS2 and 4-HNE mRNA and proteins, which explain the mechanism of the age-related mortality of Cre recombinase mice. Our results suggest that, using MHC Cre recombinase to conditionally delete genes of interest, researchers should consider the cardiotoxicity caused by Cre recombinase in the heart. For more consideration, this cardiomyopathy and the associated alterations experienced by αMHC-Cre mouse suggest that this Cre mice could be used as a model not only for classic study of the effects of genetic manipulation, but also for pharmacology investigation of human heart failure. Treatment of heart failure patients by using inhibitors or agonists of the appropriate molecules could develop a potential therapeutic approach.

## Figures and Tables

**Figure 1 ijms-24-03094-f001:**
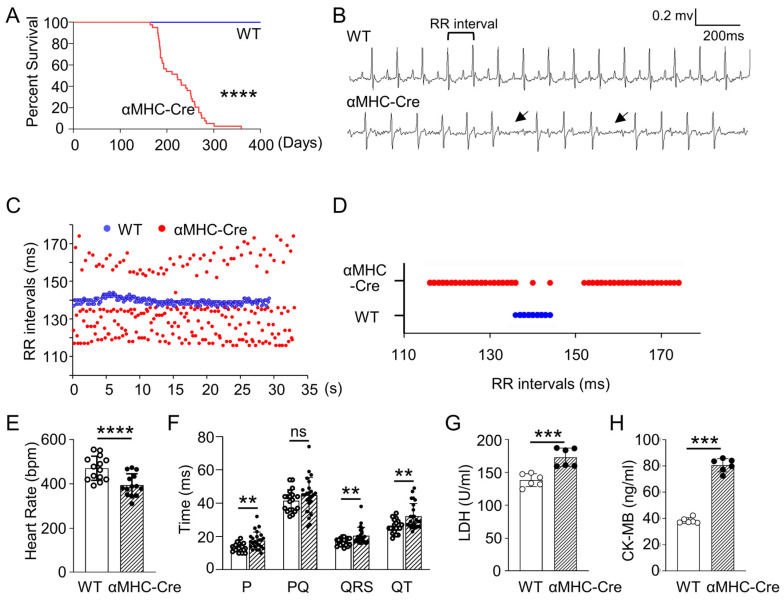
αMHC-Cre mice displayed progressive death accompanied with arrhythmia and cardiomyopathy at six months of age. (**A**) Kaplan–Meier survival curve analysis of αMHC-Cre compared to WT mice. αMHC-Cre mice began dying at 6 months, dropping off intermittently until 12 months of age (*n* = 19 mice per group). (**B**) Representative surface ECG signals were recorded from WT and αMHC-Cre mice at the age of six months. The ECG from the αMHC-Cre mice showed longer 2nd degree AV block (arrow). (**C**,**D**) Representative electrocardiograms detected from αMHC-Cre mouse showed disordered RR intervals compared to WT mice. (**E**,**F**) Heart rate (**E**) and the ECG parameters (**F**) obtained in WT (*n* = 19) and αMHC-Cre mice (*n* = 24). (**G**,**H**) Serum levels of LDH (**G**) and CK-MB (**H**) were measured in WT and αMHC-Cre mice at 6 months of age (*n* = 6 mice per group). RR interval: the time between QRS complexes. In panels (**C**,**E**,**F**), each dot represents one individual. Statistical significance was determined using two-tailed Student’s *t*-tests. ns: no significant. ** *p* < 0.01, *** *p* < 0.001, **** *p* < 0.0001.

**Figure 2 ijms-24-03094-f002:**
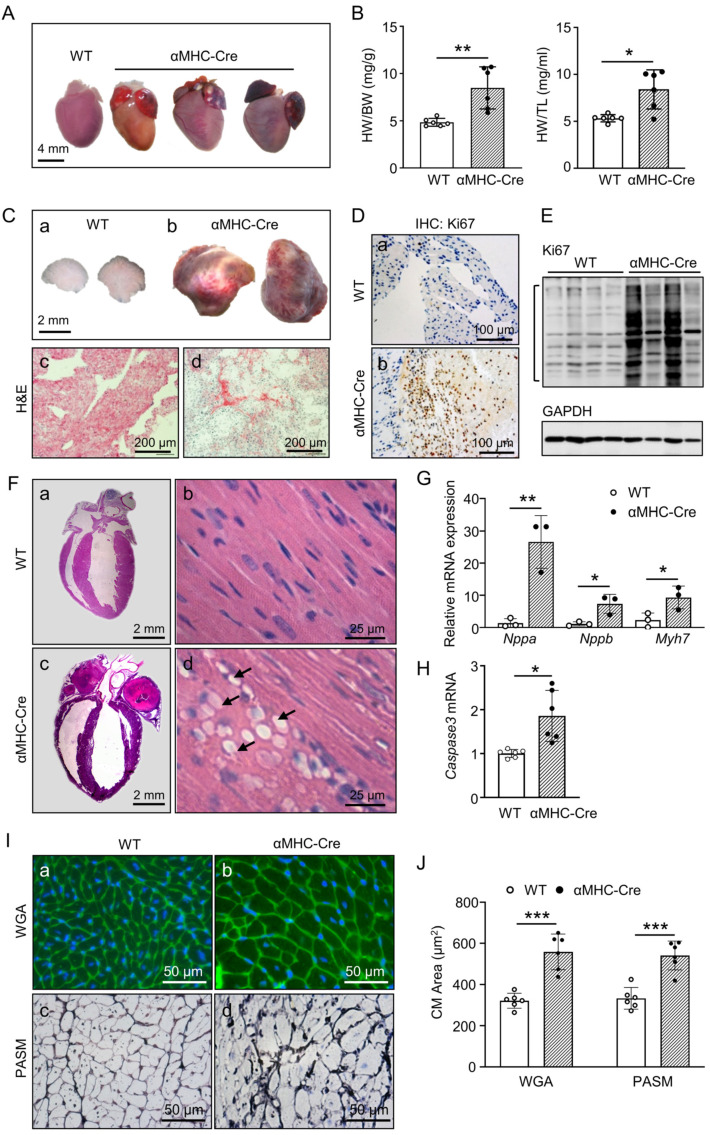
Tumor-like atrium and cardiomyocyte death lead to cardiac remodeling and heart failure in αMHC-Cre mice at six months of age. (**A**) Representative morphological images of the heart from WT and αMHC-Cre mice at six months of age. Scale bar, 4 mm. (**B**) Summary of the HW/BW and HW/TL ratio (n = 6 mice per group). (**C**) Representative left atrium morphological images were obtained from WT (**a**) and αMHC-Cre mice at six months of age (**b**) and sections stained with H&E (**c**,**d**). (**D**) Representative images of Ki67-stained left atrium sections from WT (**a**) and αMHC-Cre (**b**) mice. (**E**) Ki67 protein level of left atrium was measured by Western blot. (**F**) Left, H&E staining of heart sections from WT (**a**) and αMHC-Cre mice at six months of age (**c**). Scale bars, 2 mm. Right, higher magnification images of left ventricular sections stained with H&E from WT (**b**) and αMHC-Cre mice at six months of age (**d**). Cytoplasmic vacuolation of myofibers was present (arrow) in left ventricular of αMHC-Cre mice (**d**). Scale bars, 25 μm. (**G**) Nppa, Nppb and Myh7 mRNA were measured by real-time PCR in left ventricle of WT and αMHC-Cre mice at six months of age. Gene expression changes were presented as a fold change relative to WT mice (n = 4 mice per group). (**H**) Left ventricle Caspase3 mRNA was measured by real-time PCR in WT and αMHC-Cre mice at six months of age. Gene expression changes were presented as a fold change relative to WT mice (n = 4 mice per group). (**I**) Representative images of WGA (**a**,**b**) and PASM staining (**c**,**d**) staining for heart cross-sections from WT and αMHC-Cre mice. (**J**) Cardiomyocyte cross-sectional areas assessed using ImageJ. For (**B**,**G**,**H**,**J**), each dot represents one individual. Statistical significance was determined using two-tailed Student’s *t*-tests. * *p* < 0.05, ** *p* < 0.01, *** *p* < 0.01.

**Figure 3 ijms-24-03094-f003:**
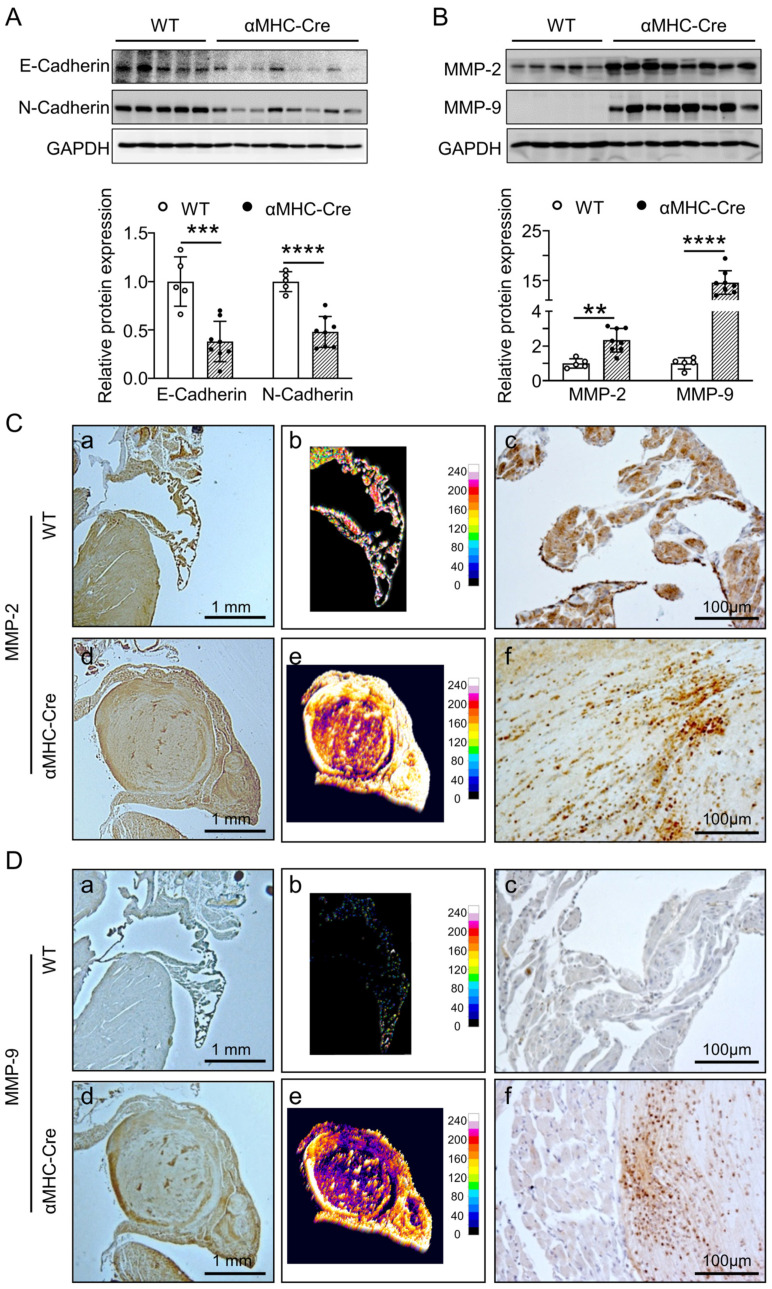
Abnormal expression of tumor-associated proteins in left atrium of αMHC-Cre mice at six months of age. (**A**) Western blots and quantification of E-Cadherin and N-Cadherin levels in left atrium (normalized to GAPDH) from six-months-old WT and αMHC-Cre mice. (**B**) Western blots and quantification of MMP-2 and MMP-9 levels in left atrium (normalized to GAPDH) from six-months-old WT and αMHC-Cre mice. (**C**) The left panel is a representative image of MMP-2 IHC staining in left atrium from WT (**a**) and αMHC-Cre (**d**) mice at six months of age. The middle panel is a 3D surface plot of MMP-2 IHC staining (**b**,**e**). The right panel is a magnified image of MMP-2 staining (**c**,**f**). (**D**) The left panel is a representative image of MMP-9 IHC staining in the left atrium from WT (**a**) and αMHC-Cre (**d**) mice at six months of age. The middle panel is a 3D surface plot of MMP-9 IHC staining (**b**,**e**). The right panel is a magnified image of MMP-9 staining (**c**,**f**). Statistical significance was determined using two-tailed Student’s *t*-tests. ** *p* < 0.01, *** *p* < 0.001, **** *p* < 0.0001.

**Figure 4 ijms-24-03094-f004:**
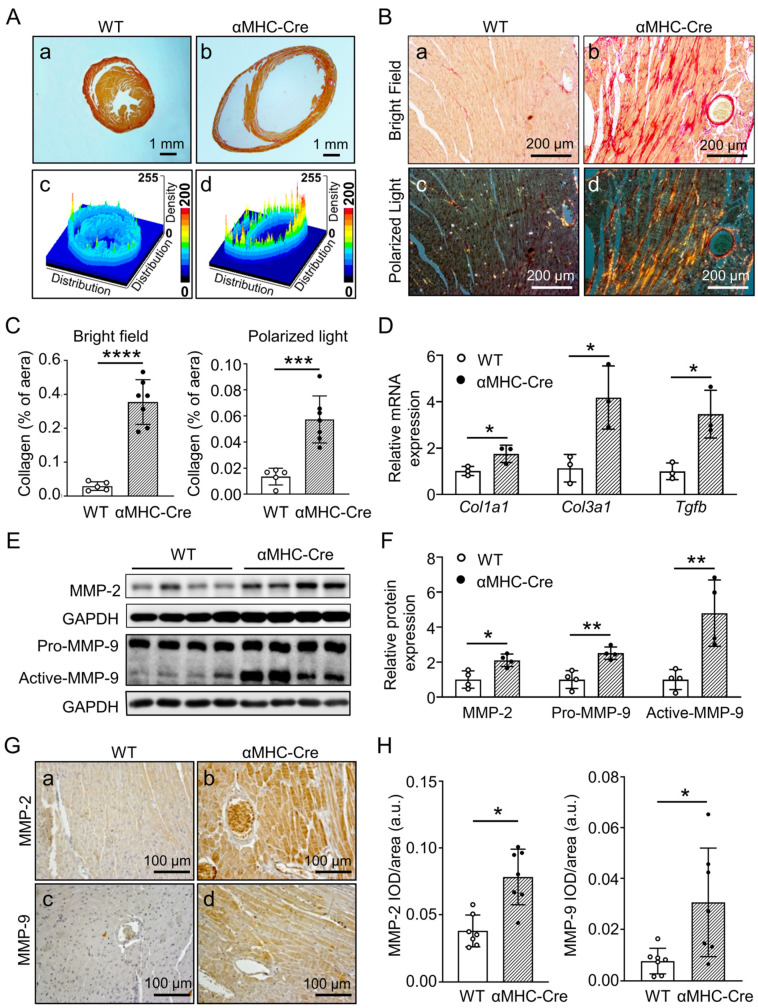
αMHC-Cre mice developed severe cardiac fibrosis with enhanced MMP-2 and MMP-9 level. (**A**) Representative Sirius red staining images of heart longitudinal sections (**a**,**b**) and their 3D surface plot (**c**,**d**) from six-months-old WT (**a**,**c**) and αMHC-Cre (**b**,**d**) mice. (**B**) Representative Sirius red-stained enlarged images using either brightfield (**a**,**b**) or polarized light microscopy (**c**,**d**) from six-months-old WT (**a**,**c**) and αMHC-Cre (**b**,**d**) mice. (**C**) Quantitative analysis of red stained area to assess fibrosis in (**B**). Quantitative analysis represents counting of multiple fields from 6 independent mice per group. (**D**) Relative mRNA levels of the cardiac fibrosis biomarkers Col1a1, Col3a1, and Tgfb in WT and αMHC-Cre mice at six months of age using real-time PCR. Gene expression changes are presented as a fold change relative to WT controls (n = 3 mice per group). (**E**,**F**) Western blots and quantification of MMP-2 and MMP-9 protein levels in left ventricle tissue (normalized to GAPDH) from six-months-old WT and αMHC-Cre mice. (**G**) Representative images of MMP-2 (**a**,**b**) and MMP-9 (**c**,**d**) immunohistochemical staining of left ventricle sections from six-months-old WT (**a**,**c**) and αMHC-Cre (**b**,**d**) mice. (**H**) MMP-2 and MMP-9 intensity was determined and quantified by Image J. For (**C**,**D**,**F**,**H**), each dot represents one individual. Statistical significance was determined using two-tailed Student’s *t*-tests. * *p* < 0.05, ** *p* < 0.01, *** *p* < 0.001, **** *p* < 0.0001.

**Figure 5 ijms-24-03094-f005:**
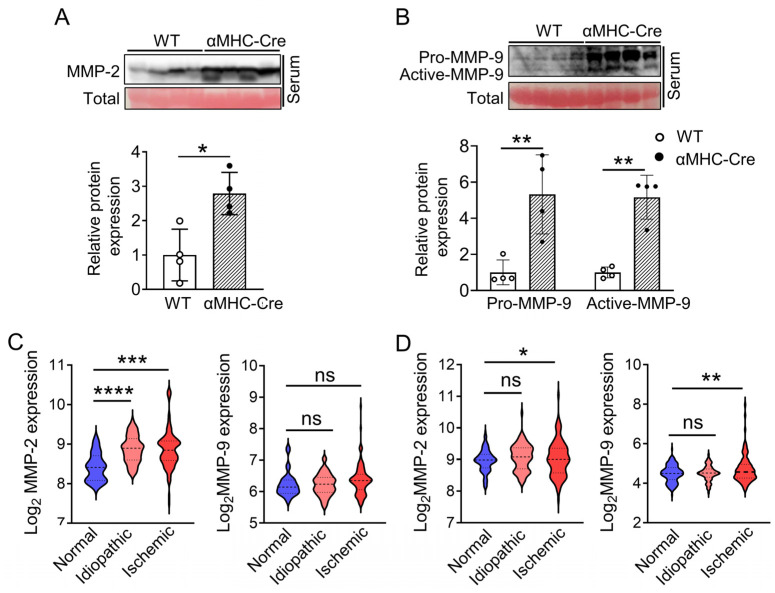
Elevation of circulating MMP-2 and MMP-9 in αMHC-Cre mice is in consistence with that of patients’ samples with heart failure. (**A**,**B**) Western blots and quantification of MMP-2 (**A**) and MMP-9 (**B**) protein levels in serum (normalized to total; Red, ponceau S staining) from 6-months-old WT and αMHC-Cre mice. (**C**,**D**) Relative MMP-2 and MMP-9 mRNA expression in human normal heart tissue or heart failure (idiopathic or ischemic cardiomyopathy) samples based on Gene Expression Omnibus (GEO) database. GEO accession number were GSE5406 (**C**) and GSE57338 (**D**). In GSE5406 dataset: n (Normal: non-failing heart) = 16; n (Idiopathic failing heart) = 86; n (Ischemic failing heart) = 108. In GSE57338 dataset: n (Normal: non-failing heart) = 136; n (Idiopathic failing heart) = 82; n (Ischemic failing heart) = 95. ns: no significant. * *p* < 0.05, ** *p* < 0.01, *** *p* < 0.001, **** *p* <0.0001.

**Figure 6 ijms-24-03094-f006:**
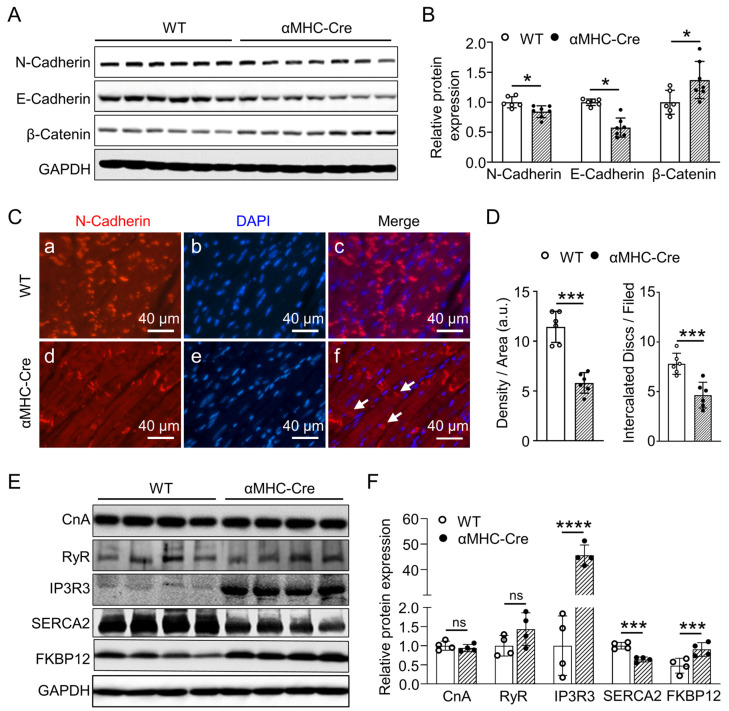
Cardiac-specific expression of Cre recombinase leads to dysregulated expression of gap junction proteins within intercalated disc and disorder of intracellular Ca^2+^ regulatory proteins. (**A**,**B**) Western blot and quantitative analysis of N-Cadherin, E-Cadherin and β-Catenin, which are components of the intercalated disc, and GAPDH as the loading control in left ventricular tissues derived from six-months-old WT and αMHC-Cre mice (n = 6–7 mice per group). (**C**) Representative immunofluorescence images of N-Cadherin (Red, **a**,**d**) localized to the intercalated disc in left ventricular from six-months-old WT (**a**–**c**) and αMHC-Cre (**d**–**f**) mice. N-Cadherin is used as an intercalated disc marker. Nuclei (**b**,**e**): blue staining. White arrow: intercalated disc. Scale bar, 40 µm. (**D**) Quantitative statistics of N-Cadherin immunofluorescence staining in D by Image J. a.u., arbitrary units. (**E**,**F**) Western blot and quantitative analysis of CnA, RyR, IP3R3, SERCA2 and FKBP12 in left ventricular derived from six-months-old WT and αMHC-Cre mice. GAPDH as the loading control (n = 4 mice per group). For (**B**,**D**,**F**), each dot represents one individual. Statistical significance was determined using two-tailed Student’s *t*-tests. ns: no significant. * *p* < 0.05, *** *p* < 0.001, **** *p* <0.0001.

**Figure 7 ijms-24-03094-f007:**
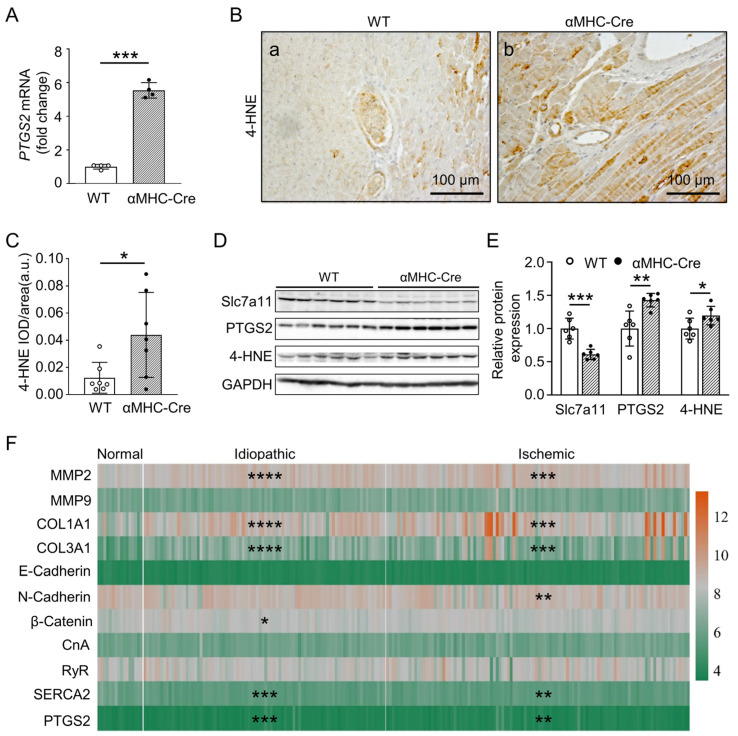
Ferroptosis signaling pathway is involved in Cre recombinase induced cardiotoxicity. (**A**) Relative levels of PTGS2 mRNA were measured in left ventricular from WT and αMHC-Cre mice at six months of age using real-time PCR. Gene expression changes are presented as a fold change relative to WT controls (n = 4 mice per group). (**B**) Representative images of 4-HNE immunohistochemical staining of left ventricular sections from six-months-old WT (**a**) and αMHC-Cre (**b**) mice. (**C**) 4-HNE intensity was determined and quantified by Image J. a.u., arbitrary units. (**D**,**E**) Western blot and quantitative analysis of Slc7a11, PTGS2 and 4-HNE, GAPDH as the loading control in left ventricular derived from six-months-old WT and αMHC-Cre mice (n = 6 mice per group). (**F**) Heat map of differential genes expression in human normal heart tissue or heart failure (idiopathic or ischemic cardiomyopathy) samples from GSE5406. n (Normal: non-failing heart) = 16; n (Idiopathic failing heart) = 86; n (Ischemic failing heart) = 108. For (**A**,**C**,**E**), each dot represents one individual. Statistical significance was determined using two-tailed Student’s *t*-tests. * *p* < 0.05, ** *p* < 0.01, *** *p* < 0.001, **** *p* < 0.0001.

## Data Availability

Publicly available datasets were analyzed in this study as described in Section 4.8.

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
