# Peer review of "Cardiac-Specific Expression of Cre Recombinase Leads to Age-Related Cardiac Dysfunction Associated with Tumor-like Growth of Atrial Cardiomyocyte and Ventricular Fibrosis and Ferroptosis"

_ijms, 2023, doi:10.3390/ijms24043094_

Round 1
Reviewer 1 Report
This paper presents an empirical study investigating cardiac-specific expression of Cre recombinase and age-related
cardiac dysfunction associated with tumor-like growth of atrial cardiomyocyte
and ventricular fibrosis and ferroptosis.
The manuscript is well written and the results of interest for both research and practice.
The paper could nicely fit into the Special Issue “Advances in Animal Models in Biomedical Research”.
Important strengths include the detailed assessments and the relevance for human medicine as well.
I am not an expert on mouse models, but I was wondering whether an RCT-like design, with randomized control groups could strengthen the claim of causality.
The conceptual contribution could be highlighted in more depth.
Which theoretical models are advanced, how in detail?
Which alternative theoretical frameworks are not supported?
Alternative explanations need to be ruled out (see also my point on control groups).
How can we make sure that there is something specific regarding the aging patterns observed?
Were there differences between male and female mice?
From a physiological perspective, such differences could be discussed in more detail.
The practical relevance could be elaborated in more detailed examples from human medicine.
Author Response
Main Comments and Suggestions:
This paper presents an empirical study investigating cardiac-specific expression of Cre recombinase and age-related cardiac dysfunction associated with tumor-like growth of atrial cardiomyocyte and ventricular fibrosis and ferroptosis.
The manuscript is well written and the results of interest for both research and practice.
The paper could nicely fit into the Special Issue “Advances in Animal Models in Biomedical Research”.
Important strengths include the detailed assessments and the relevance for human medicine as well.
Author response to reviewer major Comments and Suggestions: We thank the reviewer for his or her comments and taking efforts and spending time to review our manuscript.
Reviewer comment 1: I am not an expert on mouse models, but I was wondering whether a RCT-like design, with randomized control groups could strengthen the claim of causality.
Author response to reviewer comment 1: We appreciate the reviewer greatly for very professional comments. Generally, RCTs are considered the reference standard for driving practice. In our study, all mice used in the experiments were raised in the same environment such as maintaining environmental temperature as 20-26 degree centigrade in four seasons, 12-12 hour day-night cycling in a day. In the transgenerational breeding of mice, a fraction of mice born in the same litter were non-Cre recombinase transgenic mice, which is equivalent to wild-type litter mate control mice. The randomized control group we used was the wild-type (non-Cre recombinase enzyme gene transgenic) mice born in the same litter, and the experimental group was the Cre recombinase transgenic mice born in the same litter. Comparing the phenotypes of these two groups of mice, our focus is the mechanism of the age-related cardiomyopathy generated by the cardiac-specific expression of Cre recombinase mice in the absence of any other extra treatment. This comparison is usually design when studying gene function, which can be sufficient for the experimental design to illustrate the causality of gene effecting on the causative phenotype. In the section of Materials and Methods (4.2), we added the following sentences in our revision: “Animal maintenance of both αMyHC-Cre mice and WT littermates were household at room temperature in range of 21-26°C and in cycling of 12-hour day-time / 12-hour night time and routine conditions in the protocol approved by the ACUC-SNNU.’’ (Page 17, Lines 561-564 in the revised manuscript)
Reviewer comment 2: The conceptual contribution could be highlighted in more depth.
Which theoretical models are advanced, how in detail?
Author response to reviewer comment 2: We appreciate the valuable comments and useful suggestions from the reviewer on our manuscript. In response to this comment, we added additional two descriptions for the conceptual contribution in DISCUSSION in our revised version. One is: “According to our findings in this study, a couple of pathways may be considered as new target to develop for the heart failure such as RyR and IP3R mediated calcium signaling, MMP-cadherin associated cardiac architecture including intercalated disks of cardiomyocytes, and collagen marked cardiac fibrosis. Beside these, we unveiled the Cre promoted ferroptosis signaling indicating with the expression of increasing PTGS2/4-HNE and decreasing Slc7a11.” (Page 17, Lines 525-530 in the revised manuscript), which is partially included in our response for Reviewer comment 6 below.
Other is: “Calcium-induced calcium release (CICR) mediated with RyR and IP3R along with their regulators such as FKBP12 and is considered as the main pathway for the excitation-contractile (EC) coupling in muscle cells including cardiomyocytes in heart [51, 97, 98]. As the RyR regulator, an immunophilin protein FKBP1A can bind to immunosuppressive drugs such as FK506, rapamycin and cyclosporin A (CsA) and interacts in the mTOR pathway, which are fundamental for CICR maintained the EC coupling in general physiological condition. In our study, the significant increase of IP3R and FKBP12 but not RyR (Figure 6E and F) indicated that Cre mediated cardiopathology did involve in the regulation of RyR activity as a calcium channel in ER although the RyR expression was not increased in the Cre cardiomyocytes. The related CICR calcium influx from the ER to cytosol could be compensated via the increased IP3R proteins (Figure 6E and F). This hypothesis of the IP3R CICR calcium influx needs more evidences in cardiomyocytes as the interaction between IP3R and FKBP12 was approved in cardiomyocytes mediated by calcineurin expression in early 1995 [99] and no further research to approve this interaction.” (Page 17, Lines 533-546 in the revised manuscript).
Reviewer comment 3: The conceptual contribution could be highlighted in more depth.
Which alternative theoretical frameworks are not supported?
Author response to reviewer comment 3: We appreciate the valuable comments and useful suggestions from the reviewer on our manuscript. We would like to express that the most our data are supportive for the current theory including Calcium-induced calcium release (CICR) mediated excitation-contraction (EC) coupling in cardiomyocytes, ECM associated cardiac architecture along within intercalated disks of cardiomyocytes, and collagen marked cardiac fibrosis. Except we believe that more supporting evidences are needed on the interaction between IP3R and FKBP12 beside a publication Cell (Ref. [99] in the revised manuscript). We described this in our revision as “The related CICR calcium influx from the ER to cytosol could be compensated via the increased IP3R proteins (Figure 6E and F). This hypothesis of the IP3R CICR calcium influx needs more evidences in cardiomyocytes as the interaction between IP3R and FKBP12 was approved in cardiomyocytes mediated by calcineurin expression in early 1995 [99] and no further research to approve this interaction.” (Page 17, Lines 542-546 in the revised manuscript).
Reviewer comment 4: Alternative explanations need to be ruled out (see also my point on control groups).
How can we make sure that there is something specific regarding the aging patterns observed?
Author response to reviewer comment 4: We greatly appreciate this reviewer for the professional comments. As we explained above that we used a RCT-like design for our study aims, in which we discovered the cardiomyopathy cardiac defects led by cardiac-specific expression of Cre recombinase (control group, see response above). Based on our systematic inspection on appearance outcome and physical behavior plus EKG inspecting on cardiac function, we can observe any difference at early stage before six months of age between the αMHC-Cre mice and wild-type mice. And this is the reason why the most cardiology researchers used this model to generate cardiac specific deletion of genes which they are interested. However, when we routinely maintained the αMHC-Cre mice, we found that the mice would be mostly died about nine months of age with significant massive growth of atrial cardiomyocyte. Therefore, we initiated our study groups and further found extremely strong static difference (Figure 1A) accompanied with arrhythmias and presented with atrioventricular (AV) conduction block (Figure 1A). And then we continuously followed the design and revealed the ventricular fibrosis and ferroptosis exhibited in this research paper. Based on our systematic investigation showed in this research work, we could not think that any other detected clue can be used to possibly explain the phenomenon. We approved that this age-related cardiopathology is caused by the cardiac-specific expression of αMHC-Cre within not in their first six months of life but in the rest of their life span before their earlier death compared to the wild type control. Accordingly, we have made some additional information showed in Results (Page 2-3, Lines 96-98; Page 3, Lines 100-101 in the revised manuscript).
Reviewer comment 5: Were there differences between male and female mice? From a physiological perspective, such differences could be discussed in more detail.
Author response to reviewer comment 5: We thank the reviewer for this specific comment. In our study, we did not find any detectable difference on phenotype between female and male mice within the Cre mouse group controlled the WT. As suggested, we added the following sentences in section DISCUSSION in our revision: “In cohort studies, some reports unveiled that man and woman are in possession of different risk at cardiovascular diseases [61-63]. Generally, females are less likely to be identified the disease because of less being prescribed with aspirin, statins, and certain blood pressure medications compared to male. A recent cohort study collected data from 2020 and 2021 in the Jazan region unveiled that, among 498 coronary artery disease (CAD) patients, 20.1 % was female (100 of 498) and 79.9 % was male (398 of 498) [61]. The genetic mechanism of the sex-bias CVD was likely in possession of its significant molecular cause related to FKBP506 binding protein 12.6 [64]. The deletion mice experienced cardiac dysfunction with cardiac arrest (Ref. [65, 66] in the revised manuscript) underwent sex dependent CVD, which could be rescued with the cardiac specific over-expression of FKBP12.6 [64]. However, in our study, no detectable different phenotype between the female and male αMHC-Cre mice. From physiological perspective, the phenomenon may suggest that the hormone has no effect on the cardiomyopathy caused by Cre recombinase, which may involve in non-ryanodine receptor associated calcium pathway because, in our case, ryanodine receptor was not regulated based on the amount alternation of the RyR in Cre mice (Figure 6 E and F).” (Page 15, Lines 415-429 in the revised manuscript).
Reviewer comment 6: The practical relevance could be elaborated in more detailed examples from human medicine.
Author response to reviewer comment 6: We thank the reviewer for the valuable suggestion. According to the comments and suggestions of the reviewer, we added the following in DISCUSSION of our revised manuscript: “In human cardiovascular diseases, the CVD patients with developing heart failure suffer severe cardiac condition, in which the heart does not inflate enough blood to systematic circulation and pulmonary circulation leading to cardiac arrest. The complexity of CVD associated heart failure needs more cellular and molecular comprehension to development more efficient strategy on treatment and diagnostics. The comprehensive understanding based on animal models combined with proficient database derived from fast increasing numbers of human patients appears a unique opportunity to expend current approach to eventually decrease the risk of death. According to our findings in this study, a couple of pathways may be considered as new target to develop for the heart failure such as RyR and IP3R mediated calcium signaling, MMP-cadherin associated cardiac architecture including intercalated disks of cardiomyocytes, and collagen marked cardiac fibrosis. Beside these, we unveiled the Cre promoted ferroptosis signaling indicating with the expression of increasing PTGS2/4-HNE and decreasing Slc7a11. This ferroptosis signaling could be considered as a unique target for developing a pharmacological approach against health challenge from heart failure.” (Page 17, Lines 518-532 in the revised manuscript).
We also added the possible application that could be develop from our data in CONCLUSION: “For more consideration, this cardiomyopathy and the associated alterations experienced by αMHC-Cre mouse suggest that this Cre mice could be used as a model not only for classic study on the effects of genetic manipulation but also for pharmacology investigation on human heart failure. Treatment of heart failure patients by using inhibitors or agonists of the appropriate molecules could develop a potential therapeutic approach.” (Page 20, Lines 714-719 in the revised manuscript).
Reviewer 2 Report
The Authors present a manuscript:" Cardiac-specific expression of Cre recombinase leads to age-related cardiac dysfunction associated with tumor-like growth of atrial cardiomyocyte and ventricular fibrosis and ferroptosis" very well documented and reporting basic information extremely interesting as etiopathogenesis of specific cardiac damages.
As Clinician my requested points are.
1. Introduction too long
2. why Discussion is positioned before Materials and Methods?
3. in the Conclusion: what type of suggestion they can give to have a potential application in the clinic even if in the future? what is the linkage between molecular biology and potential application in patients?
Author Response
Main Comments and Suggestions:
The Authors present a manuscript:" Cardiac-specific expression of Cre recombinase leads to age-related cardiac dysfunction associated with tumor-like growth of atrial cardiomyocyte and ventricular fibrosis and ferroptosis" very well documented and reporting basic information extremely interesting as etiopathogenesis of specific cardiac damages.
Author response to reviewer comment: We appreciate the reviewer for taking efforts and spending time to review our manuscript.
As Clinician my requested points are.
Reviewer comment 1. Introduction too long
Author response to reviewer comment 1: We appreciate the reviewer’s constructive recommendations, and have followed her/his comments to make substantial changes in the INTRODUCTION. Please see the INTRODUCTION section in our revised manuscript. The introduction has been shortened with deletions of three paragraphs (2, 3, 4 paragraphs of the INTRODUCTION, Page 2-3, Lines 70-102 in the first submitted version).
Reviewer comment 2. Why Discussion is positioned before Materials and Methods?
Author response to reviewer comment 2: We appreciate the reviewer for this suggestion so carefully. Based on the writing order template given by the IJMS to submit our manuscript, IJMS authors are asked to format our submission in this positioning order. This requirement can be seen IJMS Microsoft Word template file (https://www.mdpi.com/files/word-templates/ijms-template.dot).
Reviewer comment 3. in the Conclusion: What type of suggestion they can give to have a potential application in the clinic even if in the future? What is the linkage between molecular biology and potential application in patients?
Author response to reviewer comment 3: We thank the reviewer for these valuable suggestions. We appreciated this suggestion and have added the following in the CONCLUSION section, “For more consideration, this cardiomyopathy and the associated alterations experienced by αMHC-Cre mouse suggest that this Cre mice could be used as a model not only for classic study on the effects of genetic manipulation but also for pharmacology investigation on human heart failure. Treatment of heart failure patients by using inhibitors or agonists of the appropriate molecules could develop a potential therapeutic approach.” (Page 20, Lines 714-719 in the revised manuscript).